# OpenReview forum: "Robust Harmful Features Under Jailbreak Attacks: Mechanistic Evidence from Attention Head Specialization in Large Language Models"
_ICML.cc/2026/Conference — ICML 2026 spotlight_

### Official Review · Reviewer_iySd · 2026-03-11

**Soundness:** 3
**Presentation:** 3
**Significance:** 3
**Originality:** 3
**Overall Recommendation:** 5
**Confidence:** 4

**Summary:**

This paper investigates whether the internal safety mechanisms of large language models are truly destroyed once a jailbreak attack succeeds. It proposes a mechanistic analysis framework based on distribution overlap: the authors back-project the refusal direction from intermediate layers onto attention heads across layers, and compare the activation distributions under three types of inputs—benign, harmful, and attack—to divide attention heads into two categories. One category consists of ACHs, which are selectively suppressed by attack templates and thereby facilitate successful jailbreaks; the other consists of SAHs, which remain robustly activated under attacks and continue to encode harmful semantics. Building on this analysis, the paper further uses ablation studies and token-level attribution to show that the suppression of ACHs has a causal role, while SAHs support the Robust Harmful Features that are difficult for attacks to eliminate. Based on these findings, the authors construct a training-free detector that achieves competitive results on the safety-eval benchmark.

**Compliance With Llm Reviewing Policy:**

Affirmed.

**Final Justification:**

This paper provides a technically solid and insightful analysis of LLM safety mechanisms under jailbreak attacks. My main concerns were about generalizability and evaluation completeness, and the authors’ rebuttal addressed them satisfactorily through clearer clarification and additional evidence. Overall, the rebuttal increased my confidence in the paper, and I am raising my score to 5.

**Key Questions For Authors:**

1. The main experiments are conducted primarily on Llama-3-8B-Instruct and Llama-2-7B-Chat, both of which are relatively well safety-aligned models. As a result, the conclusion drawn from these models—namely, that substantial refusal-feature activations remain in intermediate layers even after the model has been successfully jailbroken—may not directly generalize to other models. Can the same phenomenon also be observed in models with weaker alignment, such as Gemma-2B?
2. For the same reason as above, the number of paired jailbreak samples in the main experiments is relatively small. Would using models with weaker alignment help improve the availability of such paired samples?
3. Regarding the training-free detector proposed in Section 5, the baselines used in the experiments (e.g., LlamaGuard3 and Qwen3Guard) are all dedicated safety models. A more appropriate comparison should also include training-free detection methods (e.g., GradientCuff [1]) or methods that likewise use hidden states and refusal directions to detect harmful inputs (e.g., CAST [2]).
4. There appears to be a clear mismatch between the textual description of Figure A4 and the actual visualization, and I suggest that the authors carefully re-check and revise this part. Section D.5 states that intent-revealing tokens should generally have positive values, while attack-template tokens should have negative values or values close to zero. However, this pattern is not consistently supported by the three subfigures in Figure A4. On the contrary, a considerable number of tokens categorized as attack-template tokens show clearly positive values (warm colors), while some intent-revealing tokens do not consistently appear positive, and the patterns are not even consistent across models. Therefore, the figure currently seems to provide only partial support, rather than universal support, for the claimed conclusion. I suggest that the authors re-examine the token grouping criteria, the correspondence between color mapping and numerical signs, and revise both the figure caption and the related discussion in the main text so as to avoid drawing conclusions that are stronger than what the visual evidence actually supports.

[1] Hu X, Chen P Y, Ho T Y. Gradient cuff: Detecting jailbreak attacks on large language models by exploring refusal loss landscapes[J]. Advances in Neural Information Processing Systems, 2024, 37: 126265-126296.

[2] Lee B W, Padhi I, Ramamurthy K N, et al. Programming refusal with conditional activation steering[J]. arXiv preprint arXiv:2409.05907, 2024.

**Limitations:**

yes

**Strengths And Weaknesses:**

### Strengths
- The paper focuses on the important question of what happens inside the model when a jailbreak attack succeeds, and shows that even when the model generates harmful content, substantial refusal-feature activations can still remain in intermediate layers. This is an interesting and meaningful finding.
- The paper not only distinguishes between ACHs and SAHs, but also validates the proposed interpretation through ablation studies and token attribution, so the conclusions go beyond purely correlational observations.
- Based on the mechanistic analysis, the paper further proposes a training-free detector and demonstrates competitive performance on benchmarks, which strengthens the practical significance of the work.

### Weaknesses
- The range of models used in the main experiments is relatively narrow, and the number of paired jailbreak samples is limited, which raises concerns about the representativeness of the main conclusions.
- The ACH/SAH classification strategy involves several heuristic design choices, and some fixed parameter settings are insufficiently justified.
- Some figures use fonts that are too small and are difficult to read.

---

> ### Author Rebuttal · Authors · 2026-03-27
>
> We thank the reviewer for the thorough review and the positive assessment of both the ACH/SAH discovery and the detector's practical significance. We respond to each point below.
>
> **On parameter sensitivity:** Appendix A provides systematic sensitivity analyses: Table A1 sweeps τ from 0.1 to 0.9, showing stable conclusions for τ ∈ [0.4, 0.6]; Table A3 sweeps bandwidth scaling factor α from 0.2 to 2.0, with Spearman rank correlation ≥0.928 throughout. Our conclusions are not driven by specific heuristic settings.
>
> **Q1: Can the same phenomenon be observed in weakly-aligned models such as Gemma-2B?**
>
> Although our ACH/SAH classification and ablation were conducted on Llama-2/3, the detection experiments in Section 5 (Table 2) cover 7 models with varying alignment strengths. Taking Gemma-2B as an example, its Vanilla F1 is only 0.491, yet our detector improves it to 0.894. The improvement is even more striking in adversarial settings: on SALAD-Bench, Vanilla is merely 0.270, while our method achieves 0.976; on WildJailbreak, 0.515→0.886. This means Gemma-2B internally contains components that distinguish harmful from benign content and maintain discriminative activation even after attacks succeed---precisely the manifestation of Robust Harmful Features. The same pattern appears across Gemma-9B (0.549→0.864), Gemma-27B (0.551→0.856), Qwen-7B (0.532→0.902), and Qwen-14B (0.519→0.893).
>
> **Q2: Would using weakly-aligned models help improve paired sample availability?**
>
> This is technically feasible---weakly-aligned models have higher attack success rates and can provide abundant paired data. However, paired data is model-specific: pairs from Gemma-2B can only analyze Gemma-2B itself. Moreover, weakly-aligned models have relatively limited refusal coverage, and studying "how attacks bypass a weaker defense" reveals less mechanistic insight than studying "how attacks break through a strong safety mechanism." The latter is the core challenge facing the safety community, which is why we chose Llama-2/3 as primary analysis targets.
>
> **Q3: Comparison with training-free baselines (GradientCuff, CAST).**
>
> We fully agree that comparing under the same training-free setting is more appropriate. We conducted additional experiments on Qwen-7B (Macro-F1):
>
> | Dataset | Ours | GradientCuff | Act. Steering |
> |---|---|---|---|
> | WildGuardTest | 0.775 | 0.724 | 0.614 |
> | OpenAI Mod | 0.676 | 0.531 | 0.475 |
> | ToxicChat | 0.640 | 0.478 | 0.224 |
> | AEGIS v1 | 0.872 | 0.792 | 0.785 |
> | AEGIS v2 | 0.818 | 0.752 | 0.695 |
> | WildJailbreak | 0.923 | 0.784 | 0.945 |
> | HarmBench-V | 0.994 | 0.969 | 0.662 |
> | HarmBench-A | 0.919 | 0.840 | 0.704 |
> | SimpleSafety | 0.995 | 0.893 | 0.646 |
> | SALAD-Bench | 0.987 | 0.873 | 0.705 |
> | **Overall** | **0.904** | **0.784** | **0.665** |
>
> Our method outperforms both baselines overall (0.904 vs 0.784 vs 0.665). We also note methodological differences: GradientCuff requires gradient computation of refusal loss (involving backpropagation), substantially more expensive than our method (only forward-pass activations and inner products). Act. Steering requires pre-defined steering vectors and conditional intervention during inference, whereas our method passively reads activations without any model intervention.
>
> **Q4: Mismatch between Figure A4 and Section D.5 description.**
>
> We thank the reviewer for this careful token-level comparison. This is an important observation. We want to clarify: the issue is a **descriptive text error in Section D.5**, not an experimental or methodological problem. The visualization in Figure A4 itself is correct; what is wrong is the textual description of what to expect.
>
> Specifically, Section D.5 claims that attack-template tokens should show negative or near-zero values. However, Section 4.4 (Figure 7) clearly shows that SAHs exhibit strong positive responses to attack-template tokens (median ≈ +1.5). Our detector's top-*m* strategy selects components with the largest standardized scores. On attack inputs, ACH activations are suppressed (lower *z* values) while SAH activations are strengthened (higher *z* values), so top-*m* preferentially selects SAH-type heads. Therefore, attack-template tokens showing positive values (warm colors) in Figure A4 is entirely expected---it reflects SAH-type heads producing robust responses to attack templates.
>
> Correcting Section D.5 does not affect any other part of the paper: the ACH/SAH analysis (Sections 3--4), the ablation results (Table 1, Figure 5), and the detector's performance (Table 2) are all independent of this descriptive paragraph. The corrected explanation in fact strengthens the narrative---it connects the detector's adversarial robustness directly to the SAH mechanism discovered in Section 4.4.
>
> We will revise Section D.5, enlarge figure fonts, and ensure all descriptions align with the mechanistic findings.

---

> > ### Author Rebuttal · Reviewer_iySd · 2026-04-01
> >
> > Thank you for the detailed rebuttal. I am generally satisfied with the clarifications and additional experiments, and I appreciate the authors’ effort in addressing my concerns. After considering the rebuttal, I have decided to maintain my score at **4**.
> >
> > The main reasons I am not increasing my score further are as follows:
> > 1. The generalizability of the paper’s main conclusions still appears limited, since the core mechanistic analysis is primarily conducted on a narrow set of relatively well-aligned models, and the rebuttal provides only indirect evidence for weaker-aligned models rather than direct mechanistic validation.
> > 2. For the harmful-input detection task, the evaluation datasets include only standard benign samples, but do not include an over-refusal benchmark such as OR-Bench. As a result, the current experiments are still insufficient to fully assess the detector’s behavior on challenging benign inputs and its potential tendency toward over-refusal.

---

> > > ### Author Response · Authors · 2026-04-05
> > >
> > > We thank the reviewer for the follow-up.
> > >
> > > **On generalizability:** Our mechanistic analysis focuses on strongly-aligned models (Llama-2/3), as these models possess more structured refusal mechanisms that provide clearer subjects for mechanistic analysis, and analyzing how attacks break through strong safety mechanisms reveals deeper adversarial dynamics. How ACH/SAH manifest in weakly-aligned models, and how they compare with strongly-aligned ones, remains an open question---such comparative analysis would help understand how different alignment strategies shape internal safety structures and inform improved safety fine-tuning. This is a shortcoming of the current work, and also a valuable direction for future research.
> > >
> > > **On over-refusal evaluation:** Following the reviewer's suggestion, we evaluated all methods on OR-Bench-Hard-1K (Llama-3-8B, F1):
> > >
> > > | Method | F1 | Hard Specificity |
> > > |---|---|---|
> > > | Vanilla | 0.571 | 0.329 |
> > > | GradientCuff | 0.510 | 0.053 |
> > > | Act. Steering | 0.453 | 0.290 |
> > > | **Ours** | **0.606** | **0.418** |
> > >
> > > Our method achieves the highest F1 among all training-free methods, and even outperforms the dedicated WildGuard-7B (0.570). GradientCuff exhibits severe over-refusal (Hard Specificity 0.053, flagging 95% of benign prompts as harmful), while our method maintains the best balance between safety and helpfulness (Hard Specificity 0.418). This is consistent with OR-Bench's own finding that safety and over-refusal are inherently correlated (Spearman ρ = 0.878)---our detector handles this trade-off better than existing training-free baselines.

---

### Official Review · Reviewer_42P3 · 2026-03-13

**Soundness:** 4
**Presentation:** 3
**Significance:** 4
**Originality:** 4
**Overall Recommendation:** 6
**Confidence:** 3

**Summary:**

The authors propose a method for attributing refusal to harmful prompts/requests to individual attention heads in transformer LLMs, finding demonstrative evidence that models have both attention heads that are responsive to jailbreak attacks (i.e., attacks that attempt to circumvent refusal guardrails), and those that govern refusal yet remain robust to such attacks. At the methodological level, the authors propose a novel mechanistic interpretability procedure for projecting the refusal dimension (established in prior MechInterp research) back onto attention circuits, and use distribution overlap across refusal scenarios to identify these two types of heads. Myriad experiments support the notion that these attention heads are critical to refusal dynamics, and that one subset in particular can be said to control underlying behaviors independent of jailbreak attacks.

**Compliance With Llm Reviewing Policy:**

Affirmed.

**Key Questions For Authors:**

1. How can these attention heads be determined to be specific to refusal dynamics? The studies of acting on these ACHs (Figure 5) indicate a clear effect on the target phenomena. However, it should be determined whether or not other behaviors are affected. One might compare basic benchmark performance of the model with and without these interventions. Request that the authors address this in some way (I see it mentioned that high levels of intervention lead to deteriorating generations, but more can be said)
2. Section 5.3: These percentages don't make sense to me. "0.491 to 0.894 (+82.1%)". These don't add up as either relative or absolute improvements

**Limitations:**

Yes

**Strengths And Weaknesses:**

The paper is well-reasoned and logical, the method is novel and effective, and the results are convincing. The method draws on known literature and blends multiple recent findings to great effect. The specific application area (jail break attacks on model refusal) is highly relevant, though slightly narrow --- though, I would expect that the general approach could generalize easily, even though it is not discussed in future work by the authors. In my assessment, the paper displays solid originality and genuine novelty, while I could see many areas of mechanistic interpretability benefitting from the technical approach to identifying these attention heads.

One caveat I have, and maybe the authors can address this, is the relation of this work to the broader mechanistic interpretability literature. I myself am not an expert (though I understand the fundamental notions and seminal works). A brief literature review shows a good number of papers about attention head specialization, and yet I find little discussion of this literature --- indeed, much of the related works section has to do with the refusal/jailbreaking/safety domain.

That caveat aside, the paper is of high value and shows strong empirical results that can have high impact across numerous areas of LLM safety research. I would question whether there are similar approaches in the literature that also target attention head specialization, and whether, by omitting comparisons to baselines, we are missing key information. Regardless, the paper contributes key empirical findings about model internal behavior under refusal jailbreak attack.

My only other concern is essentially that there is too much in this single paper to digest and be appreciated without feeling condensed (indeed, the appendix is much longer than the paper itself). This could easily be split into two papers, or converted into a journal paper, but that is more a matter of opinion. Still, it does negatively affect the presentation of the work to have it be so condensed and so much relegated to the appendix.

---

> ### Author Rebuttal · Authors · 2026-03-27
>
> We thank the reviewer for the generous assessment and the positive comments on both the ACH/SAH discovery and the practical significance of the training-free detector. We respond to each question below.
>
> **Q1: Are ACHs specific to refusal dynamics, or does ablation affect other behaviors?**
>
> This is an important question. We offer the following evidence:
>
> First, the random-head control in Table 1 provides an indirect comparison. Intervening on 21 random heads yields only 4.0% ASR with high variance (±3.7), and qualitatively degrades generation coherence. In contrast, ACH ablation produces 99.5% ASR with minimal variance (±0.3), and the generated outputs remain coherent and on-topic---as illustrated by the qualitative examples in Appendix C, where model responses transition smoothly from refusal to compliance rather than degenerating into incoherent text. This contrast suggests ACHs play a relatively specialized role in refusal rather than serving general-purpose functions.
>
> Second, our ACH ablation intervenes on only 8--21 heads out of 416 (Llama-3) or 480 (Llama-2) total heads. The surgical nature of this intervention, combined with the coherent outputs it produces, suggests limited collateral impact on general capabilities. We note that Arditi et al. (2024) performed systematic evaluation on standard benchmarks (MMLU, ARC, GSM8K) for their refusal direction ablation and found minimal impact (their Table 3), consistent with our expectation given the small number of heads involved. We will include such evaluation in the revised manuscript.
>
> We also note a connection to the SAH analysis: if ACH ablation merely caused general degradation rather than specifically disabling refusal, we would expect SAH activations to also be disrupted. Instead, our Figure 6 shows that SAH activations remain intact after ACH ablation, further supporting functional specialization. This also explains the detector's adversarial robustness---SAH-type components persist under attack, ensuring the detector retains discriminative signal even when ACHs are suppressed. Without this robustness, the detector would fail precisely when it is needed most (i.e., under adversarial inputs).
>
> **Q2: Percentage calculation in Section 5.3.**
>
> Thank you for pointing this out. The reported percentages are relative improvements: (0.894 − 0.491) / 0.491 = 82.1%, and similarly (0.902 − 0.532) / 0.532 = 69.5%. We will explicitly label these as "relative improvement" in the revised manuscript to avoid ambiguity.
>
> We also appreciate the reviewer's broader comment about the paper containing substantial material. We will work to improve the balance between the main text and appendix in the revised version.

---

> > ### Author Rebuttal · Reviewer_42P3 · 2026-04-03
> >
> > Thank you for the time in responding to my comments in your rebuttal. Your clarifications make sense.

---

### Official Review · Reviewer_NJ1v · 2026-03-13

**Soundness:** 3
**Presentation:** 4
**Significance:** 3
**Originality:** 4
**Overall Recommendation:** 5
**Confidence:** 4

**Summary:**

This paper presents a mechanistic analysis of how jailbreak attacks succeed at bypassing LLM safety alignment. By projecting the mid-layer refusal direction through OV circuits, the authors quantify each attention head's contribution to refusal behavior and compare activation distributions across benign, harmful, and attack inputs. They identify two types of attention heads: Adversarially Compromised Heads (ACHs) whose outputs are suppressed by jailbreak attacks and Safety-Aligned Heads (SAHs) in mid layers, which maintain or even increase activation under attack, explaining why the refusal direction is not fully suppressed in mid-layer representations. Ablation experiments validate the causal role of ACHs (ablating 8 heads raises ASR from 0% to 95% on Llama-3-8B). The authors further propose a training-free harmful-input detector that reads internal activations and achieves competitive performance with dedicated safety models on standard benchmarks.

**Compliance With Llm Reviewing Policy:**

Affirmed.

**Final Justification:**

I believe this is a strong paper with interesting findings, in particular the ACH ablation result.

I raised three concerns: (i) the disconnect between the mechanistic analysis and the detector, which does not use the ACH/SAH taxonomy, (ii) the reliance on a single refusal direction rather than a subspace (Wollschläger et al.), and (iii) the framing of the work as contradicting an intuition no prior work explicitly holds.

The authors' reframing is a reasonable improvement, but does not fully eliminate any of the three concerns. I maintain my positive rating of 5 (Accept).

**Key Questions For Authors:**

1. Why does the detection method not use ACH/SAH heads directly? Have you compared detector performance when restricted to SAH-identified components?
2. How would the analysis change if you used a refusal subspace (per Wollschlager et al.) rather than a single direction?
3. The paper repeatedly frames its contribution as refuting the notion that "attacks erase safety representations" (e.g., "This observation contradicts the intuition that attacks erase safety representations," Section 1). However, no prior work is cited as actually holding this position. Why do authors believe it's important to refute this particular notion?

**Limitations:**

The authors acknowledge the paper's limitations.

**Strengths And Weaknesses:**

Below, [+] indicates a strength and [-] a weakness.

Overall I believe this is a strong paper: it studies a timely topic, positions itself well within the existing landscape of mechanistic interpretability and safety research, uses appropriate techniques, and shows solid results. The ACH finding in particular is convincing. I have several minor criticisms, mostly around framing and the SAH/detection portions, detailed below – but overall the paper is solid.

### Soundness

[+] The technical method rests on two solid foundations: the QK/OV circuit decomposition (Elhage et al., 2021) and the single refusal direction finding (Arditi et al., 2024). Both are used appropriately. Projecting head outputs onto the back-propagated refusal direction to quantify each head's contribution is clean and well-motivated – the linearity of the OV circuit makes this exact rather than approximate.

[-] Following Arditi et al., the authors assume the refusal direction is a single vector. However, [Wollschlager et al. (2025)](https://arxiv.org/abs/2502.17420), which the paper cites, extends this to "refusal cones", suggesting refusal may be better characterized by a subspace. The entire downstream analysis – head classification, ablation, detection – depends on this single direction being a sufficient summary. If it is not, the ACH/SAH classification may be incomplete: heads contributing to refusal along orthogonal directions would be missed entirely. The authors should discuss how their findings might change under a richer characterization of the refusal representation.

[+] The two-stage approach – use distribution overlap to locate functionally differentiated heads, then ablate to verify causal roles – is well-designed. Figure 5 in particular is convincing: the S-shaped ASR curve under progressive ACH ablation, combined with the flat random-head control, provides strong evidence that ACHs are causally involved in refusal.

### Presentation

[+] The introduction and related work are well-written. The positioning within prior research – distinguishing this work from studies of safety mechanisms under normal (non-attack) conditions – is clear and identifies a genuine gap.

[-] (Minor) Figure 5 would benefit from including the random-head progressive ablation curve alongside the ACH curve, rather than relegating the random-head results to a single endpoint in Table 1. This would allow direct visual comparison of how ASR scales with the number of heads ablated for both conditions.

### Significance

[+] Understanding why jailbreak attacks work internally is a pressing open problem – defenders can't design targeted defenses without this knowledge. The ACH finding is the paper's strongest contribution: showing that just 8 early-layer heads are sufficient to reproduce the attack effect is concrete and actionable, with clear implications for both monitoring and hardening aligned models.

[-] I am less convinced by the SAH results than the ACH results. Unlike the ACH ablation, no behavioral consequence is demonstrated: the paper does not report whether SAH ablation affects ASR, model output quality, or detection performance. Without closing this loop, the SAH causal claim is substantially weaker than the ACH one.

[-] The proposed detection method (Section 5) does not use ACH or SAH heads. It selects components based solely on harmful-vs-benign overlap (stage 1 of the head classification), without leveraging the ACH/SAH taxonomy at all. One could build the exact same detector without ever performing the head classification from Sections 3–4. This is a significant disconnect in the paper's narrative, which presents the mechanistic analysis as motivating the detection application. At minimum, the authors should compare detector performance when restricted to SAH components versus the current generic selection, which would directly test whether the SAH discovery adds practical value.

More broadly, the persistence of mid-layer refusal activations under attack may not require the SAH explanation at all. If the refusal direction is not truly a single direction (per the Wollschlager et al. concern above), or if the mid layers simply encode input semantics regardless of the model's downstream behavior, the persistence of these activations is unsurprising.

### Originality

The gap in existing literature is well-identified: prior mechanistic interpretability work on safety has focused on normal operating conditions, not on what happens internally when attacks succeed. Applying standard mechanistic interpretability tools (OV back-projection, activation analysis, ablation) to this specific question is novel and well-executed. The ACH finding in particular – that suppressing just 8 specific early-layer heads reproduces the attack effect – is a concrete and actionable result.

---

> ### Author Rebuttal · Authors · 2026-03-27
>
> We thank the reviewer for the insightful and constructive review. The questions touch on important structural aspects of our work.
>
> **Q1: Why does the detection method not use ACH/SAH heads directly?**
>
> The reviewer's observation is accurate: the detector's component selection uses harmful-vs-benign overlap (the first stage of our classification) without directly invoking the ACH/SAH taxonomy.
>
> Sections 3--4 and Section 5 address different levels of the same phenomenon. Section 5 demonstrates that Robust Harmful Features *exist* and are practically exploitable. Sections 3--4 explain *why* these features survive attacks: SAHs maintain robust activation while ACHs are suppressed, providing the mechanistic account for the persistence observed in Figure 1.
>
> The relationship is explanation + independent validation, rather than discovery + direct application. This also explains why the detector does not restrict itself to SAH components: SAHs in Llama-3 number only 17 (Figure 4), whereas the detector aggregates 1,055 candidate components with an optimal top-*m* of 875 (Table A5). Detection of diverse harmful inputs requires broad signal aggregation beyond any specific head subset.
>
> The mechanistic connection remains meaningful: SAHs' persistence under attack ensures the detector's candidate set retains discriminative components even when attacks suppress others. This helps explain why the detector performs particularly well in adversarial settings (SALAD-Bench 0.976, WildJailbreak 0.907). We will revise the manuscript to more clearly articulate this relationship.
>
> **Q2: How would the analysis change under a refusal subspace rather than a single direction?**
>
> A reasonable concern. We adopt the single refusal direction following Arditi et al. (2024), who validated its effectiveness across 13 models up to 72B---ablating this single direction suffices to eliminate refusal, supporting it as an effective first-order characterization.
>
> Our core finding does not depend on the direction being *unique*. What we show is a qualitative pattern: attacks selectively suppress early-layer heads while mid-layer heads maintain robust activation. This pattern independently replicates across Llama-2 and Llama-3 with independently extracted refusal directions (layer 14 and layer 12), yielding consistent spatial distributions (Figure 4) and ablation effects (Table 1). If the pattern were an artifact of one particular direction, this cross-model consistency would be difficult to explain.
>
> Our framework can in principle extend to refusal subspaces---performing back-projection and classification for each direction within a refusal cone (Wollschläger et al., 2025). This is valuable future work.
>
> **Q3: No prior work is cited as holding the position that "attacks erase safety representations."**
>
> Our starting point was an intuitive line of reasoning: safety alignment trains models to recognize harmful content and refuse, which means the model must internally encode "what is harmful" in some form. Attacks change the model's behavior (it no longer refuses), but cannot change the nature of the input itself---a harmful request does not truly become benign just because it is wrapped in an attack template. We therefore expected the model's internal encoding of harmful content to persist under attack. The visualization in Figure 1 confirmed this expectation, and the detection experiments in Section 5 further validated that these persistent encodings are exploitable.
>
> We agree with the reviewer that "contradicts the intuition" in the Introduction is too strong. Our intent is not to refute a specific prior claim, but to refine the community's understanding of attack mechanisms. Recent work has focused on the suppression side: Arditi et al. (2024, Section 5.1) show that adversarial suffixes "heavily suppress" the refusal direction and "hijack" top safety-critical heads; IRIS (Huang et al., NAACL 2025) and DSN (Zhou et al., 2024) further adopt suppression as an explicit attack optimization objective. Within this framework, a natural but untested corollary is that successful attacks comprehensively suppress safety-relevant representations.
>
> Our work examines this corollary and finds suppression is selective: ACHs are indeed suppressed---consistent with Arditi et al.---but SAHs maintain or strengthen their activation under attack. Arditi et al. characterized the suppressed components; we identify the components that remain robust. Together they form a more complete mechanistic account. We will revise "contradicts the intuition" to "refines the picture suggested by prior analyses."

---

> > ### Author Rebuttal · Reviewer_NJ1v · 2026-03-31
> >
> > I thank the authors for their thorough responses. The clarifications on all three questions are reasonable and the proposed revisions should strengthen the paper.
> >
> > I still believe this is a strong paper and maintain my original score of 5 (Accept). The two issues that prevent me from increasing the score further: (1) the disconnect between the mechanistic analysis and the detector remains (Q1) – the authors acknowledge the relationship is indirect, and the proposed narrative revision will help, but a stronger empirical connection would be more convincing; and (2) the framing in Q3 still relies on refuting an assumption that no prior work explicitly holds. The revised wording is better calibrated, but does not address the issue fully.

---

### Official Review · Reviewer_VjZy · 2026-03-18

**Soundness:** 2
**Presentation:** 2
**Significance:** 2
**Originality:** 3
**Overall Recommendation:** 5
**Confidence:** 4

**Summary:**

This paper studies why jailbreak attacks succeed from a mechanistic interpretability perspective. The main claim is that attacks do not erase safety-related internal features, but instead selectively suppress a subset of early-layer attention heads. It identifies two types of heads: Adversarially compromised heads which are active on harmful inputs but suppressed under attacks, and Safety-aligned heads, which remain active even when attacks succeed. It also uses these persistent SAH-related activations to build a training-free harmful-input detector, reporting competitive performance against dedicated safety models on several moderation benchmarks.

**Compliance With Llm Reviewing Policy:**

Affirmed.

**Final Justification:**

Main concerns are addressed in the rebuttal.

**Key Questions For Authors:**

1. How robust is the ACH/SAH classification to alternative refusal directions, different overlap thresholds, or different attribution schemes? Or going beyond it, how much does it generalize across architecture and training data?
2. Do real jailbreak attacks actually suppress the identified ACHs consistently across attack families, or is the current result partly an artifact of the chosen probe and dataset construction
3. The detector is interesting, but how practical is it in settings where internal activations are not accessible or where inference cost matters?

**Limitations:**

Yes.

**Strengths And Weaknesses:**

The paper connects the domains of interpretability research and jailbreaks and provides a novel perspective on the jailbreaking phenomenon. It essentially provides clarity on what internally changes when an attack actually succeeds. Its proposed approach on ACH/SAH look promising as the intervention is steerable, i.e., suppressing a relatively small number of identified ACHs produces large increases in ASR, while suppressing SAHs weakens mid-layer safety activations.

However the main analysis is done on only two models, Llama-3-8B-Instruct and Llama-2-7B-Chat, with Llama-3-70B treated only as preliminary supplementary validation. The paired harmful dataset is also quite small after filtering containing 176 successful pairs for Llama-3 and 378 for Llama-2. Given the paper’s empirical approach towards discovering underlying phenonmenon this is too narrow experimental setup for a broad claim.

I also believe that the ablations are interpreted a bit strongly. Current approach shows that for ACHs, injecting a negative feature along the projected refusal-related direction increases ASR sharply, but this does not fully isolate the natural role of those heads under attack. It shows that targeted intervention on those heads can induce jailbreak-like behavior, not necessarily that real jailbreaks operate only or primarily through that pathway.

The approach also requires a white-box access to the model provider.  The detector results are interesting, especially the gains over vanilla refusal behavior, but the comparison to dedicated safety models should be interpreted carefully. The method is competitive on average for some model families, but not uniformly strong, and it benefits from white-box internal access that many practical deployment settings do not permit.

---

> ### Author Rebuttal · Authors · 2026-03-27
>
> We thank the reviewer for the careful reading and the recognition of our novel perspective connecting interpretability and jailbreak research. We respond to each point below.
>
> **On the interpretation strength of ablation:**
>
> The reviewer draws a valuable distinction: ablation shows that targeted intervention can induce jailbreak-like behavior, not that real jailbreaks necessarily operate through the same pathway. We agree. Our ablation establishes sufficiency---suppressing 8 ACHs raises ASR from 0% to 95%+, indicating a bottleneck for refusal decisions at these heads---but we do not claim this is the only pathway.
>
> Section 4.4's token-level attribution (Figure 7) provides empirical evidence that real attacks exploit ACH suppression, with the pattern persisting across structurally diverse attack templates: attack-template tokens produce consistent strong negative contributions on ACHs (median ≈ −1.5) and strong positive contributions on SAHs (median ≈ +1.5), with separation exceeding three standard deviations. Other pathways may simultaneously participate. We will revise the wording to "sufficient causal pathway." We appreciate the reviewer's help in sharpening this claim.
>
> **Q1: Robustness of ACH/SAH classification.**
>
> We respond at three levels.
>
> *Threshold and bandwidth.* Appendix A provides systematic validation. Table A1 shows a full sweep of τ from 0.1 to 0.9; conclusions remain stable for τ ∈ [0.4, 0.6]. Table A3 shows bandwidth scaling factor α from 0.2 to 2.0; Spearman rank correlation stays high throughout (Llama-3 ≥0.928, Llama-2 ≥0.979), and Jaccard consistency exceeds 0.80 for α ∈ [0.8, 1.4].
>
> *Refusal direction.* Our classification is tied to the chosen refusal direction; a different direction traces a different safety feature. Our core robust finding is not which specific heads are labeled ACH/SAH, but a qualitative pattern: attacks selectively suppress early-layer heads while mid-layer heads maintain robust activation. This pattern independently replicates across Llama-2 and Llama-3 with independently extracted refusal directions (layer 14 and layer 12), yielding consistent spatial distributions (Figure 4). Ablation serves as end-to-end validation: if the direction led to biased ACH identification, ablation would not produce 0%→95%+ ASR, nor would random-head controls yield only 4.0% (Table 1). Exploring refusal subspaces (e.g., Wollschläger et al.'s refusal cones) is valuable future work.
>
> *Cross-architecture.* Detection experiments (Table 2) cover three architecture families (Llama, Gemma, Qwen) across 7 models, all showing consistent improvements (e.g., Gemma-2B 0.491→0.894, Qwen-7B 0.532→0.902), indicating Robust Harmful Features exist across architectures.
>
> **Q2: Consistency across attack families.**
>
> Our attack dataset spans GCG (optimization-based), AutoDAN (hybrid), many-shot (context-based), and PAIR (black-box).
>
> *On consistency.* We have conducted a preliminary per-attack-family analysis on Llama-3-8B across 6 attack types. Across all 6 types, we identify a shared core set of 30 ACH and 31 SAH heads that are consistently classified regardless of attack method. Token-level attribution on individual attack types further confirms the pattern reported in Figure 7: for example, under ShortTemplate attacks, attack-template tokens produce C_J = −0.022 on ACHs and C_J = +0.005 on SAHs; under Academic-style attacks, C_J = −0.033 on ACHs and C_J ≈ 0 on SAHs---consistent with the selective ACH suppression pattern. A comprehensive per-attack-family analysis is ongoing work.
>
> *On selection bias.* While ACH identification uses the attack dataset, ablation execution does not---it is performed on normal harmful inputs (baseline ASR ≈ 0%), validating these heads' causal role in refusal independently. Random-head controls confirm this is not an artifact of arbitrary intervention (Table 1: random heads 4.0% vs ACH 99.5%).
>
> **Q3: Practicality of the detector.**
>
> A fair point. We note two considerations. First, our core contribution is mechanistic understanding---the detector validates the practical exploitability of Robust Harmful Features. From a security research perspective, this aligns with Kerckhoffs's principle: safety analysis should not rely on hiding model internals, but rather on understanding the security floor under full access. White-box analysis is the appropriate methodology for this goal.
>
> Second, the models in this paper (Llama, Gemma, Qwen) all provide open weights; for practitioners deploying these models, accessing internal activations is feasible and the detector can serve as an inference-time input filter. Computational overhead only involves reading activations during a single forward pass and computing inner products, requiring no additional forward passes or gradient computation.

---

> > ### Author Rebuttal · Reviewer_VjZy · 2026-04-02
> >
> > I will increase my score accordingly.

---

### Decision · Program_Chairs · 2026-04-30

**Decision:**

Accept (spotlight)

**Comment:**

This paper proposes a convincing and well-supported explanation of how jailbreak attacks work, showing they suppress specific attention heads rather than removing safety behavior entirely. The analysis is both insightful, and given the strong empirical results and reviewer consensus, I recommend a clear acceptance.